# Jasmonate Signaling Pathway Modulates Plant Defense, Growth, and Their Trade-Offs

**DOI:** 10.3390/ijms23073945

**Published:** 2022-04-01

**Authors:** Cong Li, Mengxi Xu, Xiang Cai, Zhigang Han, Jinping Si, Donghong Chen

**Affiliations:** State Key Laboratory of Subtropical Silviculture, College of Forestry and Biotechnology, Zhejiang A & F University, Hangzhou 311300, China; z1143059094@163.com (M.X.); sakura13588289023@163.com (X.C.); hanzg@zafu.edu.cn (Z.H.); lssjp@163.com (J.S.)

**Keywords:** jasmonate, evolutionary origin, termination mechanism, growth, defense, growth-defense trade-off

## Abstract

Lipid-derived jasmonates (JAs) play a crucial role in a variety of plant development and defense mechanisms. In recent years, significant progress has been made toward understanding the JA signaling pathway. In this review, we discuss JA biosynthesis, as well as its core signaling pathway, termination mechanisms, and the evolutionary origin of JA signaling. JA regulates not only plant regeneration, reproductive growth, and vegetative growth but also the responses of plants to stresses, including pathogen as well as virus infection, herbivore attack, and abiotic stresses. We also focus on the JA signaling pathway, considering its crosstalk with the gibberellin (GA), auxin, and phytochrome signaling pathways for mediation of the trade-offs between growth and defense. In summary, JA signals regulate multiple outputs of plant defense and growth and act to balance growth and defense in order to adapt to complex environments.

## 1. Introduction

The plant lipid-derived hormone jasmonic acid (JA), a critical mediator of the plant defense response, is an important regulator of plant growth and development. It was in the 1980s that the first physiological processes caused by JA or methyl jasmonate (MeJA) were described, such as growth inhibition and senescence promotion [1,2]. In the early 1990s, researchers found that JA induced the accumulation of vegetative storage proteins (VSPs) after wounding in soybean leaves [3] and proteinase inhibitors (PIs) in tomato plants injured by herbivores. Turning to recent years, JA has become a research hotspot due to its diverse, complex, and specific functions.

During the last decades, much work has been performed to investigate the structure, function, and regulation of the enzymes involved in JAs biosynthesis, primarily contributing to explaining how JA is accumulated. An appropriate level of JA is essential for its biological functions [4,5,6,7], while excessive JA accumulation would trigger an overactivation of the defense machinery, which in turn comes at the expense of plant growth. Thus, a negative feedback regulation mechanism is required to delay the termination of JA signaling [8,9,10]. What has been proven beyond dispute is that plants have evolved strategies to balance both growth and reproduction with the need for defense in order to optimize their fitness in diverse environments [11]. In fact, JA modulates a major branch in the growth-defense balance [12,13,14], as a key hub of crosstalk between and with the salicylic acid (SA), ethylene (ET), gibberellin (GA), and phytochrome signaling pathways, among others [15,16,17]. Through these sophisticated JA-mediated mechanisms, plants have evolved to relocate nutrition in response to diverse and complex environmental conditions; as such, exploitation of these mechanisms may enrich both plant breeding and engineering strategies for maximizing plant fitness as well as yield.

In this paper, we review and summarize the biosynthetic pathway of JAs, a core signaling pathway, its evolutionary origin, and the multiple terminal mechanisms of JA signals, as well as the research progress considering JA in plant growth and development, stress tolerance, and recent mechanistic insights into the close relationship between growth and immunity, in order to explain and explore the possibility of maximizing the growth-defense balance.

## 2. Biosynthesis of Jasmonate

JAs are members of the family of oxylipins. There are at least three JA biosynthesis pathways in plants (Figure 1), either initiating with α-linolenic acid (shorthand notation, 18:3), starting from 7, 10, 13-hexadecatrienoic acid (shorthand notation, 16:3), or through an OPR3-independent pathway [8,18,19].

The first step of the 18:3 biosynthetic pathway occurs at the chloroplast membranes (Figure 1) [8,20]. Galactolipids release α-linolenic acid (18:3) by lipases [21]. Then, 13-lipoxygenase (LOX) catalyzes 18:3 to 13(S)-hydroperoxyoctadecatrienoic acid (13-HPOT). The coupled dehydration-cyclization of 13-HPOT, promoted by allene oxide synthase (AOS) and allene oxide cyclase (AOC), then forms 12-oxophytodienoic acid (OPDA). OPDA is transported from chloroplasts to the peroxisomes by the chloroplast envelope-localized transporter JASSY [22], peroxisomal ABC-transporter1 (PXA1), and COMATOSE1 (CTS1) [23], and is catalyzed by OPDA reductase 3 (OPR3) to 3-oxo-2-(20(Z)-pentenyl)-cyclopentane-1-octanoic acid (OPC8:0), which yields JA after undergoing three cycles of β-oxidation [24]. In a parallel pathway, dinor-OPDA (dnOPDA) is produced by the consecutive action of LOX, AOS, and AOC from 16:3 in chloroplast membranes (Figure 1) [25]. dnOPDA is transported into the peroxisome and reduced to hexanoic acid (OPC6:0) by OPR3, following which JA is generated by two cycles of β-oxidation. Recent studies have found an alternative pathway (OPR3-independent) in Arabidopsis (Figure 1) [19,26]. It is the same as the 18:3 biosynthetic pathway in chloroplast membranes, but in the peroxisome, OPDA yields dnOPDA through a single round of β-oxidation. Then, dnOPDA converts to 4, 5-didehydro-JA (4, 5-ddh-JA) in two further rounds of β-oxidation, which is subsequently reduced to JA by OPR2 in cytosol. After that, Jasmonoyl-isoleucine synthetase (JAR1) conjugates Ile to JA to form the receptor-active ligand (3R, 7S)-jasmonoyl-L-isoleucine (JA-Ile) [19].

## 3. Jasmonate Signaling Pathway

### 3.1. The Core JA-Ile Pathway

The core jasmonate signaling pathway consists of interconnected functional modules that regulate the transcriptional state of JA-responsive genes. The bioactive JA-Ile is perceived by cognate nuclear receptors CORONATINE INSENSITIVE 1 (COI1). As a typical F-box protein, COI1 first binds to SKP1-LIKE proteins (ASK1 or ASK2) and subsequently forms the Skp1/Cul1/F-box complex (SCF^COI1^) with Arabidopsis Cullin1 to recruit JASMONATE-ZIM DOMAIN (JAZ) proteins for ubiquitination and degradation [24,27,28]. JAZ proteins are significant repressors that connect early responsive MYCs (MYC2/3/4/5) transcription factors (TFs) and, later, responsive downstream genes [8,29]. Actually, 1 COI, 13 JAZ, and 4 MYC proteins had been described in Arabidopsis [30,31,32]. In the absence of biotic effectors in plants, JA-Ile is typically at a lower level, and the JAZ family proteins accumulate, which physically bind to and inhibit MYCs [33,34] through two distinct mechanisms [29,33,35]. First, MYC-bound JAZ proteins recruit the co-repressors TOPLESS (TPL), either directly by ETHYLENE-RESPONSE FACTOR-associated amphiphilic repression (EAR) motifs located at the N terminus of a subset of JAZ proteins (JAZ5/6/7/8/13), or indirectly through NOVEL INTERACTOR OF JAZ (NINJA), an EAR motif-containing protein [36,37,38] (Figure 1). Constitutively, it prevents MYCs from binding to the G-box sequence of downstream gene promoters, thereby blocking the JA signaling pathway [9]. JA-Ile is rapidly activated in response to various stresses, followed by the generation of the JAZ-JA-IIe-SCF^COI1^ E3 protein complex, which ubiquitinates and degrades JAZs proteins [33]. After the degradation of JAZ proteins, TPL is disintegrated to release MYCs, inducing the transcription of the responsive genes in the JA signaling pathway [39].

In contrast to vascular plants, the bioactive COI1-JAZ ligand in *Marchantia polymorpha* is not JA-Ile, but rather dnOPDA -the precursor of JA-Ile. Both the 18:3 and 16:3 pathways in chloroplasts are the same as in vascular plants, whereas OPDA produced by the 18:3 pathway in the peroxisome passes through one round of β-oxidation to dnOPDA. In the nucleus, the receptor COI1 directly recognizes dnOPDA and exerts its functional activity (Figure 1). The ligand specificity of *M. polymorpha* is due to a single residue substitution in COI1, which explains the evolutionary events between the bryophytes and vascular plants [40,41]. The *Mpcoi1* mutation is functionally complemented by AtCOI1 and gains JA-Ile responsiveness [40]. Similarly, in contrast to all of the embryophyta studied so far, the *M. polymorpha* genome has single JAZ and MYC orthologs (Figure 1), which are functionally conserved in other land plants [40,41,42]. *Mpjaz* mutants show similar developmental phenotypes to Arabidopsis JAZ-depleted mutants (Table 1), such as growth inhibition, rhizoid emergence, gemma cup formation, cell size, or antheridiophore development [13,42,43], and the defects of *Mpjaz-1* mutant are complemented by overexpression of *AtJAZ3* [42]. As in Arabidopsis (Table 1), JAZ deficiency in *Marchantia* compromises fertility, perhaps as a consequence of constitutive activation of MYCs along with MYCs interacting with MED25 to activate JA-responsive genes, which requires further confirmation [12,13,44]. Similar to AtJAZs, for the proper regulation of JAZ function, MpJAZ transcripts exist splicing (elimination of the Jas domain, *M**pJAZ∆Jas*), which is stabilized against hormone-induced degradation and responsible for terminating the JA response [42,45]. The trade-off between overactivation of jasmonate signaling and reproductive fitness is widely conserved, suggesting that the JA core signaling pathway has been maintained since plants first colonized terrestrial habitats.

### 3.2. The Termination of JA Signaling

The JA signaling pathway is terminated to create time-delayed negative feedback regulation mechanisms, which serve to ensure sufficient initiation of the JA signal [45].

In varying stress responses and different growth stages of plants, JA is metabolized into active, partially active, or inactive components through different approaches to balance the homeostasis of JA, including conjugation, hydrogenation, carboxylation, decarboxylation, methylation, esterification, sulfonation, glycosylation, and the formation of 12-OH-JA lactone [8]. JA-signaled transcriptional responses are tightly integrated with the accumulation of JA and JA-Ile [9]. A key feature of JA-Ile, as the molecular trigger of signaling, is its rapid accumulation induced by herbivores, wounding, necrotrophic pathogens, or other types of stress [100]. Furthermore, JA-Ile promotes the expression of two kinds of COI-dependent JA-Ile deactivated enzymes: CYP94 oxidase and amidohydrolases, including IAR3, ILL5, and ILL6, and so on (Figure 2A) [8]. First, JAR1 transforms JA into JA-Ile. Then JA-Ile is oxidized twice by CYP94 P450 family enzymes to 12-OH-JA-Ile and 12-COOH-JA-Ile in turn. Next, JA-Ile and 12-OH-JA-Ile generate JA and 12-OH-JA through the amidohydrolases IAR3, ILL5, and ILL6 [101]. This process usually leads to a decrease in JA-Ile. In addition, JA can be catalyzed by jasmonic acid carboxyl methyltransferase (JMT) to produce MeJA [8,102]. In reality, plant growth, including seeding growth and hypocotyl elongation, under warm temperature conditions requires tight control of available JA-Ile levels, mainly through JASMONATE-INDUCED OXYGENASES (JOXs) and ST2A-mediated JA catabolism [103,104,105]. This co-operative action of metabolic and catabolic enzymes results in a highly dynamic balance of JA and JA-Ile associated with the corresponding biological responses [106].

The COI1 protein is strictly regulated by a dynamic balance of SCF^COI1^-mediated stabilization and 26S proteasome-mediated degradation (Figure 2B). The stable formation of SCF^COI1^ and degradation of excessive COI1 maintain the stable content of COI1 in plants, which is of great significance for biological functions [107].

COI1-dependent removal of pre-existing JAZ proteins can provide for rapid activation of defenses and other JA-dependent processes, whereas rapid synthesis of new JAZ proteins ensures attenuation of the JA response soon after transmission of the signal [29]. Arabidopsis group IV JAZ proteins (AtJAZ7, AtJAZ8, and AtJAZ13), a particular sub-family of JAZs, harbors a divergent Jas motif that interacts weakly with COI1 [31]. Although the atypical ZIM domain of JAZ7, JAZ8, and JAZ13 fails to mediate the interaction with NINJA, they have a conserved EAR domain that directly interacts with the co-repressor TPL [31,107,108]. Following their interaction with MYC2 through the Jas-like motif, they behave as a constitutive repressor in the JA pathway (Figure 2C) [37,107,108]. For example, overexpression of *JAZ13* confers JA-insensitivity and decreased resistance to insect herbivory [108]. JAZ10.4, a variable splice of JAZ10, lacks the Jas motif and mediates desensitization to JA through interacting with MYC2, MYC3, and MYC4 TFs through the CMID region, which can form extensive contact with the transcriptional activation domain (TAD) of MYCs (Figure 2C) [45]. Arabidopsis plants overexpressing JAZ10.4 were found to be insensitive to exogenous JA application and exhibited high tolerance to JA-induced degradation [79]. The inert JAZ proteins and alternative splicing of JAZ10 pre-RNA create part of the regulatory circuit to attenuate JA responses. It is intriguing that plants have developed a mechanism to prevent excessive desensitization to JA responses mediated by JAZ splice variants [63]. Following the generation of JAZ splice variants depending on the mediator subunit MED25, MED25 recruited the splicing factors RPR39a and PRP40a, which promoted the correct splicing of JAZ genes (full splicing of Jas intron) and prevented the overproduction of JAZ splice variants (Figure 2C).

Arabidopsis bHLH subclade IIId proteins, including bHLH17/JAM1, bHLH13/JAM2, bHLH3/JAM3, and bHLH14, compete with MYC2-like TFs to bind to the G-box elements of its target gene promoters and impair the formation of the MYC2-MED25 complex, thereby deactivating MYC2-like TFs-dependent gene transcription (Figure 2D) [9,109,110], which has also been proved in tomato MTB (MYC2-targeted bHLH) TFs, homologs of Arabidopsis JAM TFs. The MID region in the TAD domain of MYC2 is essential for the interaction between MYC2 and MED25. Tomato MTB proteins lack a canonical MED25-interacting domain due to an altered MID domain (AMID) and therefore fail to interact with MED25. Moreover, the AMID plays a significant role in the MTB1-JAZ interaction, which possibly helps to repress gene expression and to terminate JA signaling [9]. As a result, MYC2 and bHLH IIId proteins generate an auto-regulatory negative feedback circuit to terminate JA signaling in a highly organized manner [9,10].

Therefore, multiple highly organized strategies have been developed to control the accurate termination of JA signaling in plants in order to avoid over-stunted growth as a result of excess defense response. Obviously, both the activation and the deactivation/desensitization of JA responses must be under tight control.

## 4. The Functions of JA in Growth and Development

### 4.1. Promotion of Plant Regeneration

Recent studies have suggested new roles for JA in promoting plant regeneration [111,112,113]. JA reduces the quiescent center (QC) quiescence in the root stem cell niche (SCN) through the RBR-SCR network and stress response protein ETHYLENE RESPONSE FACTOR115 (ERF115) [113]. Furthermore, JAs serve as wound signals during de novo root regeneration (DNRR) by activating ERF109 (Figure 3). On the one hand, ERF109 co-operates with SET DOMAIN GROUP8 (SDG8)-mediated histone H3 lysine 36 trimethylation (H3K36me3) to upregulate *ANTHRANILATE SYNTHASE α1* (*ASA1*), a tryptophan biosynthesis gene in the auxin production pathway and promotes cell fate transition to form the root primordium [5,112]. On the other hand, JA-induced ERF109 transcription stimulates CDK interactor *CYCLIND6;1* (*CYCD6;1*) expression, functions upstream of *ERF115* and promotes regeneration [111,113]. Therefore, the JA tissue damage response pathway induces stem cell activation and regeneration, as well as activating growth after environmental stress.

### 4.2. Regulation of Reproductive Growth

JA has been found to coordinate stamen filament elongation, anther dehiscence, and pollen viability [6]. Male sterility is recognized as one of the typical characteristics of JA disability. The Arabidopsis JA biosynthesis mutant *opr3*, as well as *aos*, showed defects in anther and pollen development resulting in male sterility (Table 1) [48,114,115]. A maize double mutant *opr7opr8* with dramatically reduced JA has displayed reproductive deficiency and strong developmental defects, which were rescued by exogenous JA (Table 1) [116]. Mutations in the Arabidopsis JA receptor COI1 (*coi1-1*) caused abnormal anthers and pollen, leading to reproductive deficiency (Table 1) [30,61,63]. *coi1-2* and *coi1-8* displayed a partial fertility phenotype [27,63]. *coi1-16* exhibited fertility in a temperature-sensitive manner [28]. Overexpression of rice *COI1* genes (either *OsCOI1a* or *OsCOI1b*) could restore the fertility of an Arabidopsis *coi1-1* mutant, while *OsCOI2* failed to [101]. Overexpression of *OsJAZ6*, which interacts with OsJAZ1, altered JA signaling and led to abnormal spikelet development [85]. However, tomato *JAI1*, a homolog of Arabidopsis *COI1*, is required for the maternal control of seed maturation. A *Sljai1* mutant displayed reduced viability of seeds as a result of a defect in female reproductive development, which was associated with the loss of accumulation of JA-regulated proteinase inhibitor proteins in reproductive tissues (Table 1) [100]. Furthermore, JA regulates petal and stamen growth by releasing the inhibitory effect of JAZ on the downstream R2R3-MYB TFs MYB21/MYB24 (Figure 3) [117]. Overexpression of *MYB21* in *coi1-1* plants restored stamen development [8,118]. MYB21 and MYB24 connect physically with MYC2, MYC3, MYC4, and MYC5 to control stamen development [8,110]. The *myc2 myc3 myc4 myc5* quadruple mutant exhibited short filament, delayed anther dehiscence, and unviable pollen grains at the floral stage (Table 1) [8], while *coi1-1* plants overexpressing the *MYC5* and *MYC3* exhibited restored stamen maturation and productivity [8,110].

JA represses the vegetative-reproductive maturation transition as well. In Arabidopsis, JA acts through COI1-JAZ/TOE-FT to inhibit flowering (Figure 3). A *coi1-2* mutant, *JAZ1Δ3A* transgenic plants, and *JAZ9* overexpression plants displayed early flowering (Table 1) [119]. JAZ proteins interact with APETALA2 (AP2) family TFs TARGET OF EAT1 (TOE1) and TOE2 and repress the transcription of *FLOWERING LOCUS T* (*FT*) [119]. In tomatoes, SlJAZ2 regulates plant morphology and accelerates flower initiation. Plants overexpressing *SlJAZ2* exhibited quicker leaf initiation, shorter plant height and internode length, earlier lateral bud emergence, and more advanced flowering transition (Table 1) [99].

### 4.3. Actions of JA in Vegetative Growth

JA plays a dual role in seed germination in co-operation with abscisic acid (ABA). In cold-stimulated germination of wheat seeds, JA content rapidly increased after up-regulation of JA biosynthesis-related gene expression and further suppressed ABA biosynthesis by repressing two key ABA biosynthesis genes, *TaNCED1* and *TaNCED2* [120]. During the rice germination period, ABA acts upstream of JA and cooperatively inhibits rice seed germination through the SAPK10-bZIP72-AOC regulation pathway. It directly activates the transcription of AOC by phosphorylating bZIP72 through SAPK10 in order to promote the biosynthesis of JA and inhibit rice seed germination [121].

JA acts through COI1 to induce leaf senescence [110]. Arabidopsis *coi1* mutants showed a stay-green phenotype under dark-induced senescence conditions (*coi1-1*) [62] and under MeJA treatment (*coi1-2*) [64], indicating that COI1 plays a role in leaf senescence (Table 1). A *jaz7-1* (WiscDsLox7H11) mutant displayed severer dark-induced leaf yellowing, as well as quicker chlorophyll degradation (Table 1) [73]. In addition, MYC2, MYC3, and MYC4 function redundantly, binding to and activating the promoter of their target gene, *SENESCENCE-ASSOCIATED GENE29* (*SAG29*), to activate JA-induced leaf senescence (Figure 3). However, MYC2/3/4-activated JA-induced leaf senescence is attenuated by the bHLH sub-group IIId factors (bHLH03/13/14/17), competitively binding to the promoter of SAG29 and repressing its expression [110]. A recent study has found that *DNA binding-with-one-finger 2.1* (*Dof2.1*), a JA-inducible gene, acts as an enhancer of JA-induced leaf senescence through the MYC2-Dof2.1-MYC2 feed-forward transcriptional loop [122]. In reality, JA signaling-mediated leaf senescence is also regulated by the circadian clock. For instance, evening complex (EC), a core component of the circadian oscillator, negatively regulates leaf senescence by directly binding the promoter of MYC2 and repressing its expression, as well as gating JA signaling in order to regulate leaf senescence [123].

As has been widely accepted, stomata are cavities surrounded by guard cells on the leaf epidermis, which regulate water balance, gas exchange, and immune response to pathogens. JA has the ability to control the opening and closing of stomata in a COI1-independent manner. *Pseudomonas syringae* pv. tomato (Pto) DC3000 is a pathogenic factor of tomato bacterial spot disease, producing coronatine (COR) as a virulent factor to activate the JA pathway, promoting the formation of JAZ2-COI1 complexes and triggering JAZ degradation through the 26S proteasome to inhibit SA-dependent defense responses to *P. syringae*, thus inducing stomata opening and facilitating the entrance of the pathogen into the leaf apoplast [67]. Consistently, the dominant mutant *SlJAZ2Δjas*, lacking the C-terminal jas domain, presents inhibiting effects toward the reopening of stomata induced by COR (Table 1) [98]. Similarly, Arabidopsis *jaz2-3* mutants are partially impaired in pathogen-induced stomatal closing and, thus, are more susceptible to *Pseudomonas* (Table 1). Remarkably, Arabidopsis dominant *jaz2∆jas* mutants are resistant to *P. syringae* but retain unaltered resistance against necrotrophs (Table 1) [67]. Furthermore, a recent study has demonstrated that AvrB, a type III effector protein of *P. syringae*, induces stomatal opening through a canonical JA signaling pathway involving COI1 and NAC TFs (Figure 3). It promotes the interaction between COI1 and JAZ by the RPM1-INTERACTING4 (RIN4)-Arabidopsis plasma membrane H+-ATPase (AHA1) pathway and induces the degradation of multiple JAZs to open stomata [124].

Trichomes are epidermal appendages with different forms, structures, and functions, such as protecting plants against herbivores. The JA/COI1 signaling pathway plays an important role in the promotion of glandular trichomes. In tomatoes, deficiencies in JA perception block glandular trichome formation [100]. The trichome-preferentially expressed SlJAZ4 is the critical component in JA-triggered tomato trichome elongation by interaction with HOMEODOMAIN PROTEIN8 (SlHD8). SlHD8 promotes elongation by activating the expression of expansin genes (*EXPs*; Figure 3) [125]. Similarly, *Artemisia annua* AaJAZ8 interacts with a positive regulatory factor, AaHD1, repressing its transcriptional activity and inhibiting the formation of glandular trichomes [126]. It is worth mentioning that JA and GA hormonal signals synergistically regulate plant development. The Arabidopsis GL3/EGL3/TT8 complex in the bHLH family binds to WD-Repeat and MYB proteins in order to form the WD-repeat/bHLH/MYB complex. Both JAZ and DELLA target the WD-repeat/bHLH/MYB complex to repress trichomes formation. The initiation of trichomes is regulated by the WD-repeat/bHLH/MYB complex in a JA-dependent manner, which was attenuated in the JA signaling-deficient mutant *coi1-1*, along with low expression of GL2 and MYB23 [127].

JA inhibits primary root growth in Arabidopsis through MYC2-mediated repression of *PLETHORA1* (*PLT1*) and *PLT2*, which are known as the key TFs of the auxin-regulated root meristem activity and maintenance [128]. It is also reported that two Arabidopsis YUCCA genes, *YUC8* and *YUC9*, which participate in auxin homeostasis and root development, are regulated by oxylipins dependent on the COI1 signal transduction pathway [129]. In fact, mutations in COI1 lead to insensitivity to JA-inhibitory primary root elongation [27]. The triple mutant *myc2 myc3 myc4* (*myc2**/3**/4*) showed an obvious reduction in JA-dependent primary root growth inhibition (Table 1) but less severe than that in *coi1-1* [32]. However, the role of JA in lateral root development is different. The exogenous application of MeJA up-regulates the expression of ERF109, which stimulates *ASA1* gene expression and increases the content of auxin in Arabidopsis, thus promoting the occurrence of lateral roots and inhibiting taproot elongation (Figure 3) [130]. The defect of lateral root formation of *asa1-1* mutants after MeJA treatment is closely related to the significant down-regulation of PIN2 protein levels by JA [131].

As described in Table 1, researchers have also uncovered other roles for JA in developmental and growth-related processes. For example, *coi1-37* displayed leaf epinasty, dark green leaves, strong apical dominance, and enhanced meristem longevity (Table 1) [66]. Overexpression of *JAZ9* in plants led to phenocopy of the *coi1* mutant with longer hypocotyls and petioles under low-in-tensity light conditions and early flowering, GA-hypersensitivity phenotype (Table 1) [29]. Over-expressed *JMT* rice plants showed reduced height and yield (Table 1) [82].

## 5. Role of JA during Plant Defense Responses

### 5.1. JA Mediates Plant Defense against Pathogens

In natural environments, plants may be infested by different types of pathogens, including biotrophic, hemi-biotrophic, and necrotrophic pathogens. Accumulating evidence points to the JA signaling pathway as mainly corresponding to plant immunity against necrotrophic fungal pathogens, including *Alternaria brassicicola*, *Botrytis cinerea*, *Plectosphaerella cucumerina*, *Pythium* spp., and so on [95,132,133]. In Arabidopsis, the JA biosynthetic mutants *fad3fad7fad8* and *jar1* showed increased susceptibility to *A. brassicicola* [46,47] and *P. irregular* [134], respectively (Table 1). The *coi1-1* mutants exhibited increased susceptibility to the necrotrophic fungi *A. brassicicola*, *B. cinerea*, and *P. cucumerina* (Table 1) [132,133]. JAZ6 has been proven to be a crucial component in time-of-day defense against *B. cinerea*. When responding to *B. cinerea*, *jaz6-1* lost the time-of-day difference due to failure to induce *JAZ6* expression and did not exhibit enhanced susceptibility at subjective night while retaining resistance at dawn (Table 1) [71]. The maize double mutant *opr7opr8* showed extreme susceptibility to a root-rotting oomycete (*Pythium* spp.) [116]. A tomato mutant (*jai1*) suffered 100% mortality from root-rot disease [95] and exhibited increased susceptibility to *B. cinerea* and *Fusarium* species (Table 1) [96,97]. A recent study has reported that WRKY75 exploits the JA signaling pathway by directly binding to downstream target genes such as ORA59 in order to positively regulate the Arabidopsis defense response against *B. cinerea* and *A. brassicicola* (Figure 4A) [135]. IQ-MOTIF-CONTAINING PROTEIN 1(AtIQM1), a Ca^2+^-independent CaMBP, increases the activity of the JA biosynthetic enzymes ACX2 and ACX3 by interacting with CATALASE2 (CAT2), thereby positively regulating JA content and the *B. cinerea* resistance of Arabidopsis [136]. In *Rosa chinensis*, upon JA treatment, free forms of RcMYB84 and RcMYB123 were released due to JAZ1 degradation, further activating the plant’s defense against *B. cinerea* [137].

Although transcriptional regulatory elements involved in the JA signaling pathway (e.g., COI1, JAZs, and MYC2) are relatively conserved in plants, the manner in which MYC2 regulates downstream genes is species-specific. Arabidopsis defenses against *B. cinerea* are regulated by the expression of two groups of genes through AtMYC2 [138,139]. The first group of genes involved in JA-mediated systemic responses to wounding is activated by AtMYC2 through the direct regulation of NAC019. The second group includes genes involved in defense against pathogens, which are negatively regulated by AtMYC2 through suppression of ERF1 (Figure 4A). Arabidopsis *myc2/jin1* mutants showed increased resistance to *B. cinerea* and *Fusarium oxysporum* but attenuated resistance to insects (Table 1). However, in tomatoes, SlMYC2 both positively regulates wounding-responsive genes by activating JA2L and pathogen-responsive genes by activating ERF.C3 [140]. A *B. cinerea* infection assay in tomatoes presented significantly larger necrotic lesions in MYC2-RNAi plants than in the wild type [140].

In Arabidopsis *ein2* or *coi1* mutant, exogenous JA and ET alone or in combination failed to induce the expression of downstream defense genes (such as *ERF1* and *PLANT DEFENSIN 1.2 (PDF1.2*)) [139,141], illustrating that these two signaling pathways are concomitantly essential for the activation of plant defense responses (Figure 4A). JA enhances the transcriptional activity of EIN3/EIL1 through the removal of JAZ proteins, which recruit RPD3-type histone deacetylase (HDA6) as a co-repressor to repress EIN3/EIL1 transcriptional activity and further activates downstream *ERF1/ORA59PDF1.2* cascades, thereby defending against necrotrophic pathogens [139,141,142]. Arabidopsis CCCH protein C3H14 regulates the activation of WRKY33-ORA59 cascades to correspondingly promote JA/ET signal transduction and camalexin biosynthesis in order to increase the plant’s tolerance to *B. cinerea* [143]. A recent study has reported that the Arabidopsis *BIG* gene orchestrates the antagonism between two parallel ERF1/ORA59 and MYC2 branches in the JA pathway that determine resistance to pathogens and wound response. BIG deficiency promotes JA-dependent gene induction and increases JA production but restricts the accumulation of both ET and SA. Eventually, JA-induced stomatal immunity is impaired after BIG disruption [16]. Moreover, JA acts in a complex signaling network combined with SA signaling pathways after *P. syringae* infection. COR produced by *P. syringae* hijacks a signaling module, COI1-JAZ2-MYC2/3/4-ANAC19/55/72, to control stomatal dynamics during the invasive process (Figure 4A). In detail, COR hijacks the JA pathway to suppress the SA pathway by directly activating the expression of SA biosynthesis enzyme inhibitor NACs TFs (ANAC19/55/72) through MYC2/3/4, which is targeted by JAZ2, thereby inhibiting the SA-mediated defense response against *P. syringae* [67]. In rice, following infection by *Magnaporthe oryzae*, the SA signaling regulator OsNPR1 sequesters OsbHLH6 in the cytosol (activating JA signaling when localized to the nucleus) and activates SA signaling but represses JA signaling to control rice resistance to *M. oryzae* [15]. The PRC1 protein LHP1 is involved in the repression of the MYC2 branch of the JA/ET pathway of immunity by inhibiting the expression of NACs and their target BSMT1 and promoting the accumulation of SA, thereby enhancing the defense against *P. syringae* (Figure 4A) [144]. The Arabidopsis C-terminal binding protein ANGUSTIFOLIA (AN) antagonistically regulates plant defense against the hemi-biotrophic pathogen *P. syringae* and the necrotrophic pathogen *B. cinerea* (Figure 4A). AN interacts with TYROSYL-DNA PHOSPHODIESTERASE1 (TDP1) and imposes transcriptional repression on MYB46, which encodes a transcriptional activator of the SA biosynthesis gene *PHENYLALANINE AMMONIA-LYASE* (*PAL*) while releasing TDP1-imposed transcriptional repression on WRKY33, a master regulator of the JA/ET signaling pathway. The antagonistic effect of MYB46 and WRKY33 through AN regulation suggests a transcriptional co-regulatory mechanism of SA and JA/ET pathways, indicating a transcriptional node regulating the trade-offs between (hemi)biotrophic and necrotrophic defenses [145].

In contrast to the biotroph/necrotroph dichotomy mentioned above, it is noteworthy that JA is also required for the induction of immunity against biotrophic and hemi-biotrophic pathogens, including the fungi *F. oxysporum* [146] and *Verticillium dahlia* [96], as well as the bacteria *Pectobacterium atrospecticum* and *Xanthomonas oryzae* pv. *oryzae* (*Xoo*) [88,147]. In rice, *OsJAZ8∆C*-overexpressing plants, which lack the Jas domain, exhibited a JA-insensitive phenotype and reduced JA-induced resistance to *Xoo* (Table 1) [88]. ABA-inducible SnRK2-type kinase SAPK10-mediated phosphorylation on Thr129 of WRKY72 weakens its DNA-binding ability to AOS1, promotes the endogenous JA level, and finally enhances *Xoo* resistance, which highlights the role of ABA-JA interplay in post-translational modification and an epigenetic regulation mechanism [147].

### 5.2. JA Acts as a Double Agent in Plant Defense against Herbivorous Insects

Plants usually have the intrinsic ability to resist attacks from herbivorous insects through a combination of constitutive and inducible defenses. JA is employed as a central defense signal for plant resistance against herbivory. Upon insect feeding, JA synthesis is rapidly triggered, inducing massive defense-related genes, the production of diverse secondary metabolites (terpenoids, phenolics, as well as nitrogenous and sulfur-containing compounds), specific defense proteins (protease inhibitors, polyphenol oxidases, leucine aminopeptidase, lectins, and chitinases), and the formation of a physical barrier (e.g., trichomes) to suppress or prevent the feeding (Figure 4B) [148,149]. Arabidopsis plants deficient in JA biosynthesis and signaling typically suffers more damage from molluscan herbivores [149]. MeJA treatment induced the expression of glucosinolate (GS) synthesis genes, as well as GS accumulation [150,151]. MYC2/3/4 directly binds to the promoters of GS biosynthetic genes and interacts with GS-related MYBs, thereby promoting the JA-mediated synthesis of secondary metabolites and defense against external assaults [76]. Consistent with this, the Arabidopsis *myc2**/3**/4* triple mutant is completely devoid of GS and is extremely susceptible to the generalist herbivore *Spodoptera littoralis* [76] and spider mite herbivory (Table 1) [77]. In rice, the concentrations of JAs were dramatically increased after a brown planthopper (BPH) attack, along with an increase in H_2_O_2_ level [82]. BPH performed better on JA-deficient lines (AOC and MYC2 knockout) than on wild-type (WT) plants due to the attenuation of defensive secondary metabolites accumulation (Table 1) [80]. Additionally, rice COI1 RNAi lines increase susceptibility to chewing insect *Cnaphalocrocis medinalis* as a result of impairing inducible defense by induction of trypsin protease inhibitor (TrypPI), peroxidase (POD), and polyphenol oxidase (PPO) [10]. Tomato plants treated with JA showed reduced numbers of *Frankliniella occidentalis* (thrips), *Helicoverpa armigera*, flea beetles, and aphids due to an increase in the activities of PIs and polyphenol oxidase [152]. Furthermore, the development of glandular trichomes in tomato leaves is controlled, in part, by the JA pathway [153], providing an important anti-insect defense layer [154]. Tomato *jai1* plants exhibited several defense-related features, including the inability to express JA-responsive genes, severely compromised resistance to two-spotted spider mites, and reduced monoterpene production due to the abnormal development of glandular trichomes (Table 1) [100]. Upon herbivore attack, JA signaling is activated, and subsequently, MYC2, MYC3, and MYC4 mediate indole-3-acetic acid (IAA) biosynthesis through the activation of *YUCCA9* expression [155,156].

As a countermeasure, many specialized herbivores manipulate their defense by modulating the JA signaling pathway. For instance, Whitefly (*Bemisia tabaci*)-induced tomato plant volatiles prime SA-dependent defenses and suppress JA-dependent defenses, thus rendering neighboring tomato plants more susceptible to whiteflies, which has been confirmed by experiments with volatiles from caterpillar-damaged and pathogen-infected plants [157]. The effector HARP1, which is released from the cotton bollworm during feeding, also attenuates the plant defense by interacting with JAZ repressors to restrain COI1-mediated JAZ degradation, therefore blocking JA signaling [158]. Interactions between brassinosteroids (BRs) and SA/JA have been reported in the insect-resistance response in rice. BR biosynthesis is induced after BPH infestation by upregulating the expression of *OsBRI1*/*OsBZR1*, followed by the JA synthesis-related gene *OsLOX1*/*OsAOS2*. The JA signal down-regulates the expression of the SA biosynthesis gene *ISOCHORISMATE SYNTHASE 1* (*OsICS1*) and *OsPAL* to inhibit the SA-mediated defense response to BPH. The suppressive effects of BRs on the SA pathway were eliminated in JA-deficient and JA-insensitive mutants. These results indicate that BRs obviously promote the susceptibility of rice host plants to BPH by modulating the SA/JA co-action defense responses [159].

### 5.3. JA Plays Vital Roles in the Arms Race between Plants and Viruses

BRs and JAs finely participated in building the plant defense system in a synergistic or antagonistic manner. The synergistic effect of BR and JA enhances the resistance to *rice stripe virus* (RSV) (Figure 4C). BR-induced RSV resistance is blocked in *osmyc2* knockout plants (Table 1). RSV reduces JA-mediated defense by increasing the accumulation of the BR signaling negative regulator OsGSK2 [86] and by making it physically interact with the JA positive regulator OsMYC2, resulting in the degradation of OsMYC2 by phosphorylation, thus promoting its infection [87]. Collectively, these results demonstrate that BRs positively contributed to regulating JA-mediated resistance. However, an antagonistic relationship between BR and JA effects in viral defense has been reported. *Rice black-streaked dwarf virus* (RBSDV)-infected rice plants show that genes of the JA pathway (LOX1, AOS2, JMT1, and MYC2) are up-regulated, while genes in the BR pathway (D11, OsDWARF4, D2, CPDs, BRI1, and BZR1) are down-regulated. The line *Go* (a mutant overexpressing OsGSK2, which can block the BR signaling pathway) plants showed a marked decrease in susceptibility to RBSDV (Figure 4C). *coi1-13* mutant infection experiments and application of exogenous hormones indicated that JA-mediated defense can suppress the BR-mediated susceptibility to RBSDV infection in a manner dependent on the JA co-receptor OsCOI1 [83]. Furthermore, ABA is also involved in the JA-mediated resistance to the virus. ABA negatively regulates rice defense against RBSDV by preventing JA-mediated accumulation of reactive oxygen species (ROS) [160]. In recent years, an increasing number of studies have shown that viruses perform host-plant manipulation on their specific host plants. Rice viruses in the genera *Fijivirus*, *Tenuivirus*, and *Cytorhabdovirus* all possess transcriptional repressors that directly disassociate the OsMED25-OsMYC3 complex, inhibit the transcriptional activation of OsMYC3, and then combine with OsJAZ proteins to cooperatively overcome the JA pathway in a manner that benefits viral infection and the feeding activity of their vectors [161]. RBSDV-encoded P5-1 regulates the ubiquitination activity of SCF E3 ligases and inhibits JA signaling to benefit its infection [162]. RSV, transmitted by the small brown planthopper (SBPH), is the type member of the genus *Tenuivirus*. MeJA treatment attracted SBPHs to feed on rice plants, where a JA-deficient mutant was less attractive than wild-type rice. This is because the JA pathway, induced by the coat protein, activates the plant defense against RSV while attracting SBPHs to feed, thus benefiting viral transmission [163].

### 5.4. JA Regulates Plant Tolerance against Abiotic Stresses

JAs have been implicated in the control of plant responses to abiotic stimuli, such as mechanical stress [35], salt [164], drought [84,88,165], UV irradiation [166], and ozone exposure [167]. JA synthesis genes, including *LOX2*, *LOX3*, *AOS*, *AOC*, and *OPR3* in Arabidopsis, and *TomLoxD* and *AOS* in tomato, are significantly up-regulated by exogenous JA and mechanical damage [168,169]. Tomato *res* mutants, which accumulate JA and show remarkable growth inhibition and important morphological alterations, can restore cell structure alterations under salty stress (Table 1) [164]. In rice, *OsJAZ1*-overexpressing plants were more sensitive to drought stress treatment, while *jaz1* mutant plants indicated increased drought tolerance (Table 1) [84]. In wheat, when under drought stress, JA acts on the upstream of ABA in response to initiating the drought tolerance of plants. Plants first promote the biosynthesis of ABA and JA and subsequently induce related signal pathway genes, such as *SnRK2* and *MYC2*, which further activate the transcriptional activities of downstream genes, including calmodulin, disease resistance protein RPM1, CAT, SOD, and HSP70 [170]. Recent evidence in apple (*Malus domestica*) has revealed that E3 ligase MdMIEL1 (MIEL1, MYB30-interacting E3 Ligase1) and MdJAZ proteins directly modulated the BBX37-ICE1-CBF module to achieve the dual regulation of JA-mediated cold stress [171].

## 6. JA Mediates the Trade-Offs between Growth and Defense

JA controls a multitude of transcriptional programs affecting plant regeneration, reproductive process, phenotype formation, and stress defense and exerts strong control over the growth-defense balance. In most cases, it contains multiple biological processes rather than a single hormone pathway to shape plant growth and their response to defenses. However, how do interactions among JA core pathway components and crosstalk between JA and other biological processes control myriad aspects of growth, development, reproduction, and immunity?

### 6.1. Interactions among JA Core Components

In Arabidopsis, the functions of MYC2/3/4, including restricting leaf and root growth, activating leaf defense, and enhancing susceptibility to the pathogen *P. syringae* and promoting defense against insect herbivory, have been validated through phenotypic comparison of the quintuple mutant *jazQ* (*JAZ1/3/4/9/10*) (Table 1) and a *jazQ myc2 myc3 myc4* octuple mutant. Therefore, MYC TFs exert epistatic control over JAZ-repressible transcriptional processes that govern JA-mediated growth-defense trade-offs [12]. The researchers further combined 13 *JAZ* gene mutants to examine the effects of long-term JAZ deletion on defense, growth, and reproductive output. The results considering an uncovered *jaz* decuple mutant (*jazD*) of 10 *JAZ* genes (*JAZ1-7, -9, -10,* and *-13*) showed that it possessed resistance to insect herbivores and fungal pathogens but had slow vegetative growth and poor fertility (Table 1). The absence of the remaining JAZ repressors in *jazD* mutant plants further aggravated growth arrest, led to almost no seed production, and even facilitated the spread of necrotic lesions and tissue death under extreme conditions [43]. Therefore, the dual role of the inhibitory effect of JA on growth and the enhancement of defense provides evidence of the antagonistic relationship between growth and immunity.

### 6.2. Crosstalk between JA and Other Phytohormones

JA is an important hormone related to plant defense, while gibberellin (GA) is an important hormone mediating plant growth. Antagonistic signal crosstalk between the binding of bioactive JAs or GAs to cognate receptors leads to proteasome-dependent degradation of JAZ/DELLA proteins that, at the resting state, represses cognate TFs involved in defense (e.g., MYCs) or growth (e.g., phytochrome-interacting factors, PIFs) (Figure 5). DELLAs, serving as central regulators linking the crosstalk between JA and GA, inhibit MYC2-JAZ interactions, thus liberating MYC2 to promote the JA response. Meanwhile, in the presence of GA, DELLAs are eliminated to compromise the JA response [172]. In rice, the JA-induced defense against rice root knot nematode (*Meloidogyne graminicola*) requires SLENDER RICE1 (SLR1, a DELLA protein in rice) accumulation to repress the GA pathway [17]. In Arabidopsis, the interaction between RGA (a DELLA protein) and PIF3 is inhibited by JAZ9, thereby promoting hypocotyl elongation. Both a *della* quintuple mutant and a *pif* quadruple mutant displayed insensitivity to JA-induced hypocotyl inhibition, indicating that the DELLA-PIF interaction is required for JA-mediated growth inhibition during the response. Additionally, overexpression of *PIF3* could partially overcome JA-induced growth inhibition. Therefore, a molecular cascade involving the COI1-JAZ-DELLA-PIF signaling module elucidates that the antagonistic effects of JA and GA reconcile the growth-defense dilemma [68].

More recent works have described the interaction between JA and auxin as of particular relevance for the control of plant growth-defense trade-offs [173,174]. Wound-inducible amidohydrolases (IAH) contribute to JA and auxin levels to coordinate stress responses and development by controlling JA-Ile and IAA contents [175]. Both JA and auxin perception depend on SCF-type ubiquitin protein ligase (E3) complexes [173,176], and both the *arx1* and *arx6* mutants result not only in reduced auxin response but also a reduction in JA sensitivity [177], which might reduce JA responsiveness through the recruitment of such shared components by auxin, thus leading to a limitation of JA-mediated defense responses and amplification of auxin-mediated growth responses, and vice versa [173,174]. Therefore, the interaction between JA and auxin contributes to the fine-tuning of plant stress responses and development.

### 6.3. Crosstalk between JA and Phytochrome Signaling Pathway

Light is one of the most significant signals for plants to respond to the external environment. JA, in conjunction with phytochrome photoreceptors, is able to affect a variety of plant growth, development, and defense processes.

In Arabidopsis, the activities of *FARRED ELONGATED HYPOCOTYLS3* (*FHY3*) and *FARRED IMPAIRED RESPONSE1* (*FAR1*) are induced by shading (low R/FR) (Figure 5). On the one hand, FHY3/FAR1 activates the expression of atypical bHLH transcriptional co-factors PHYTOCHROME RAPIDLY REGULATED (PAR1) and PAR2, which inhibit the expression of downstream growth-related genes by forming heterodimers with PIF. On the other hand, FHY3/FAR1 and MYC2 jointly promote the expression of defense-related genes. In these two processes, JAZ proteins inhibit the activities of FHY3 and MYC2 from maintaining the balance between growth and defense [178]. Recent evidence has revealed that antagonistic crosstalk between JA and the red-light receptor phytochrome B (phyB) participates in the plant growth-defense balance. phyB mutation completely rescued the growth and reproductive defects in a *jazQ* mutant without affecting the defense level (Figure 5) [13]. Uncoupling of growth-defense antagonism in *jazQ phyB* plants has been attributed, in part, to simultaneous activation of the MYC and PIF modules [13] which, in wild-type plants, antagonizes one another partly through the interaction between JAZs and DELLAs [40]. However, unlike the *jazQ phyB* mutant, which both grows and defends, *jazD phyB* plants maintained the strong defense of the *jazD* but showed weak growth status, which reveals an independent pathway of phyB for the defense-related growth restriction. Moreover, the slow growth of *jazD* and *jazD phyB* plants is tightly correlated with up-regulation of the Trp biosynthetic pathway, together with enhanced expression of genes encoding enzymes for the conversion of Trp to defensive GSs [14,76]. In this case, it is possible that the growth-defense trade-offs do not rely only on transcriptional networks but also depend on strong metabolic constraints due to the reallocation of metabolites for defense [179].

## 7. Conclusions and Perspectives

JA has various physiological effects. As a signal molecule, on the one hand, it plays a key role in plant growth and development, including plant regeneration, anther and pollen development, flowering time, seed germination, leaf senescence, stomata closure, and trichome development [73,98,113,118,119,121,125]. On the other hand, it also provides plants with a strong defensive capability to ward off the majority of their natural enemies, including necrotrophic pathogens [139], herbivorous insects [80], and viruses [160]. Intriguingly, some cunning enemies-particularly herbivorous insects [159] and viruses [83], have evolved ingenious mechanisms to hijack the JA signal network in order to suppress or evade host defense responses.

Although a host of studies in Arabidopsis and tomato have described the synthesis and regulation mechanisms of JA, there are still many problems to be solved, especially with regard to the complexity of transcriptional regulation of JA as a key role in co-regulation with other pathways. Unlike in higher plants, there are single COI1, JAZ, and MYCs orthologs in the liverwort *M. polymorpha* (Figure 1), which allows it to serve as a window to overcome some bottlenecks caused by genetic redundancy in vascular plants, as well as to unveil the evolutionary history of JAs in growth and defense. Recent evidence has supported the dn-OPDA (Figure 1), instead of JA-Ile, as the bioactive COI1-JAZ ligand in *M. polymorpha* [40]] and the function MpJAZ is conserved when compared with Arabidopsis [42]. However, JAZ-interacting TFs have not been studied in *M. polymorpha*. Exploration of the epistatic interactions within the COI1-JAZ-TFs module in *M. polymorpha*, instead of formidable members of the AtJAZ and AtMYC families in Arabidopsis, is expected to open new frontiers in the field of JA biology.

When plants are in defensive status, their growth will be inhibited, which is generally believed to be due to the transfer of resources for growth to the synthesis of defensive metabolites [11]. Previous evidence has demonstrated that the constitutive activation of jasmonate-mediated defenses can be achieved with minimal effects on growth [180,181,182]. Increased resistance to both necrotrophic fungi and herbivorous insects but unaffected growth status has been detected in Arabidopsis when after down-regulation of the JASMONATE-ASSOCIATED VQ MOTIF 1 (JAV1) repressor [180]. In addition to genetic strategies, it may be possible to use chemical tools or endophytes to break the antagonistic relationship between growth and defense. For instance, NOPh (a phenyloxime derivative of COR stereoisomer)-treated Arabidopsis showed a moderate defense response to necrotrophic pathogens without growth inhibition through selective activation of the ERF-ORA branch by binding with co-receptor COI1-JAZ9 [181]. *Epichloë* fungal endophytes in plants alleviate the trade-off between growth and defense by regulating GA, auxins, SA, or JA pathways [182]; however, the associated molecular basis remains to be determined. The uncoupling of growth-defense trade-offs has been observed in *jazQ phyB* plants but not *jazD phyB* plants [13,14], suggesting that the balance of growth and defense not only involves the interaction between JA and phytochrome signaling pathways but also that with other pathways. Additional research is required in order to determine how the complex networks between the JA signaling pathway and other pathways act to synergistically regulate in order to uncouple, or, at least, minimize, such trade-offs to produce plants with robust growth and defense simultaneously.

Previous efforts have provided several meaningful clues that JAs can induce the biosynthesis of terpenes, flavonoids, and other medicinally active ingredients, thus crucially contributing to plant secondary metabolism. The JAs-mediated biosynthesis of secondary metabolites is mainly modulated by the SCF^COI1^ complex, JAZ protein, MYC2, and other TFs, such as WRKYs and MYBs [145,147], which activate or inhibit the expression of multiple important enzyme genes in plant secondary metabolic biosynthetic pathways by interacting with cis-acting elements in their promoters. For example, exogenous MeJA treatment of the hairy roots of *Salvia miltiorrhiza* induced the biosynthesis of tanshinone, phenolic acids, and other active ingredients by regulating the expression of the secondary metabolite synthesis gene *SmWRKYs* [183]. Therefore, there exists a bridge between TFs, biosynthetic genes, and secondary metabolites. However, in most medicinal plants that are rich in metabolites, the JA-mediated biosynthetic pathways of secondary metabolites have not yet been clearly elucidated. It is imperative that further study clarifying the molecular mechanism of JAs regulating the synthesis of secondary metabolites in medicinal herbs be carried out, which will broaden our knowledge of the functions and signaling events associated with JAs, and which may even benefit human health.

## Figures and Tables

**Figure 1 ijms-23-03945-f001:**
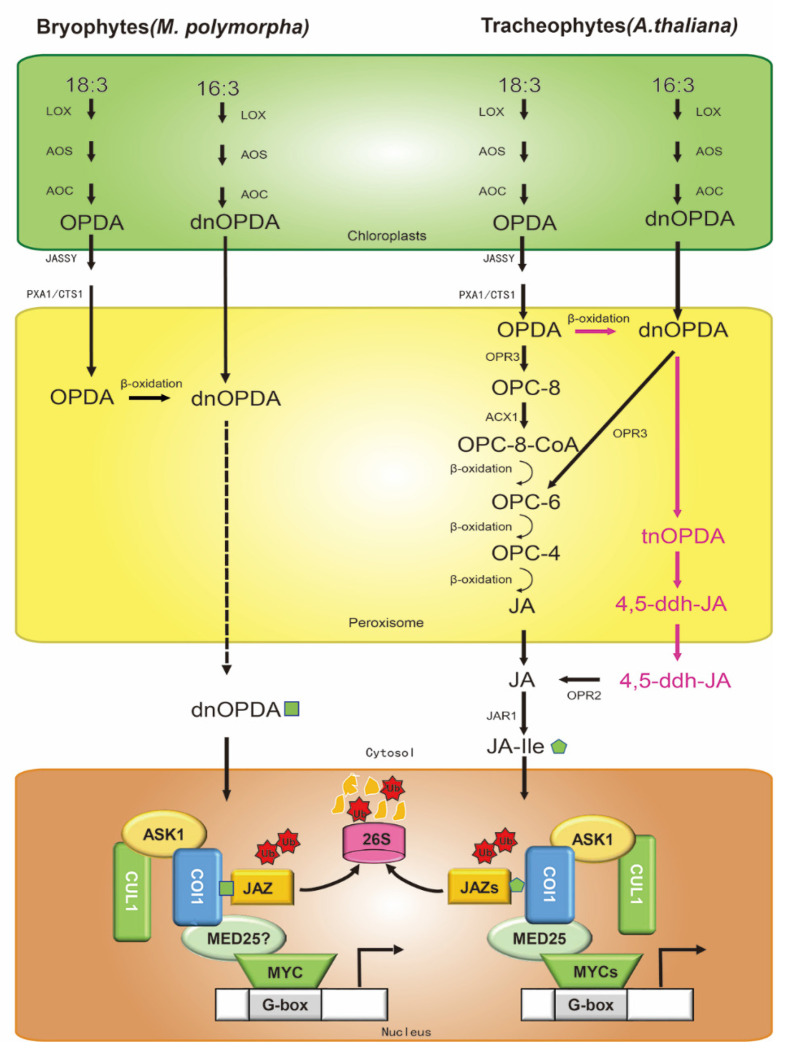
Biosynthesis and core signaling pathways of jasmonic acid (JA) in Arabidopsis and *Marchantia polymorpha*. There are three synthetic pathways in Arabidopsis (right): (1) the 18:3 pathway; (2) the 16:3 pathway; and (3) the independent OPR3 pathway, in which the cytoplasm JAR1 catalyzes JA to form JA-Ile with biological activity. JA-Ile interacts with the COI1-JAZ complex to degrade JAZ(s) by ubiquitination through the 26S proteasome degradation pathway. The difference in jasmonic acid evolution between *Marchantia polymorpha* (left) and Arabidopsis is mainly reflected in (1) the number of *JAZ* and *MYC* genes in *M. polymorpha* and Arabidopsis; (2) the bioactive COI1-JAZ ligand is dnOPDA in *M. polymorpha*, but JA-Ile in Arabidopsis. The mechanisms of MYC and MED25 in the JA signaling pathway in *M. polymorpha* are not proven (question mark). Ub, ubiquitin. Arrows: activations; bar-headed arrows: repressions.

**Figure 2 ijms-23-03945-f002:**
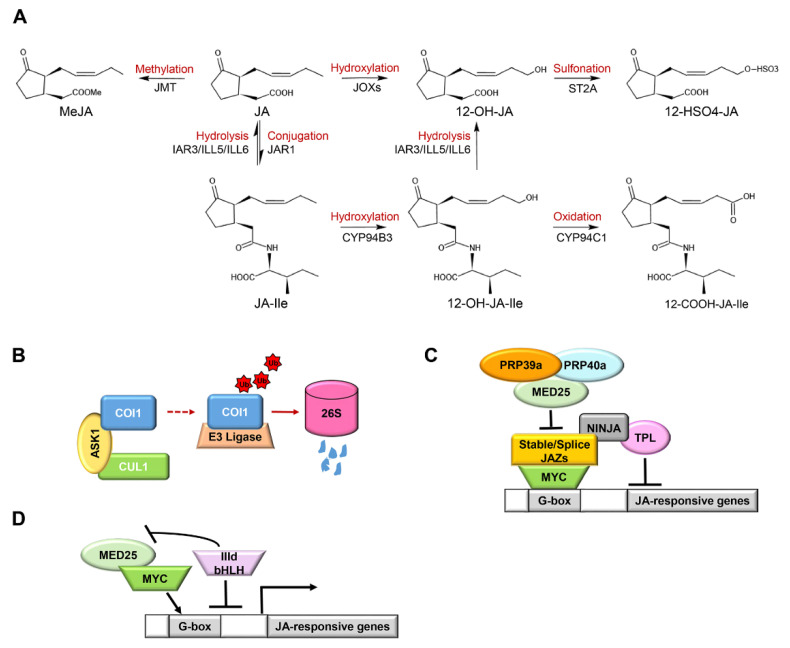
The terminal regulation mechanism of jasmonic acid (JA) signaling: (**A**) the dynamic banlance of JAs. Metabolic and catabolic enzymes maintain a dynamic balance between JA and JA-Ile. Methyl jasmonate (MeJA) is formed by JA catalyzed by jasmonic acid carboxyl methyltransferase (JMT). JAR1 transforms JA into JA-Ile. JA-Ile is continuously oxidized twice by CYP94 P450 family enzymes to generate 12-OH-JA-Ile and 12-COOH-JA-Ile in turn. The amide bond of JA-Ile/12-OH-JA-Ile is broken by amide hydrolases (IAR3/ILL5/ILL6) to generate JA/12-OH-JA, respectively. JA can be directly hydroxylated to 12-OH-JA and further sulfated to form 12-HSO4-JA; (**B**) COI1 degradation pattern. Jasmonate receptor COI1 forms the SCF^COI1^ complex with ASK1 and CUL1. Excessive COI1 protein is recruited and degraded by ubiquitination through the 26S proteasome pathway; (**C**) the inert and alternative splicing of JAZ-mediated JA attenuation pattern. JA signal suppressor JAZ protein can bind to NINJA protein and directly or indirectly recruit co-inhibitor TPL to inhibit the transcription of MYC2, thereby inhibiting the expression of JA-responsive genes. Inert JAZs (JAZ7, JAZ8, and JAZ13) have a conserved EAR domain that directly interacts with TPL. JAZ10.4, a variable splice of JAZ10, lacks the Jas motif and mediates desensitization to JA. MED25-recruited splicing factors PRP39a and PRP40a promote the correct splicing of *JAZ* genes and prevent the overproduction of splicing variants, thereby regulating the activation of JA signal; (**D**) bHLH-like protein-mediated JA attenuation pattern. bHLH IIId (JAM1/2/3, bHLH14) proteins compete with MYC-like TFs to bind the G-box element on the promoter of the JA response gene, inhibit the formation of the MYC2-MED25 complex, and negatively regulate the JA response. Arrows: activations; bar-headed arrows: repressions.

**Figure 3 ijms-23-03945-f003:**
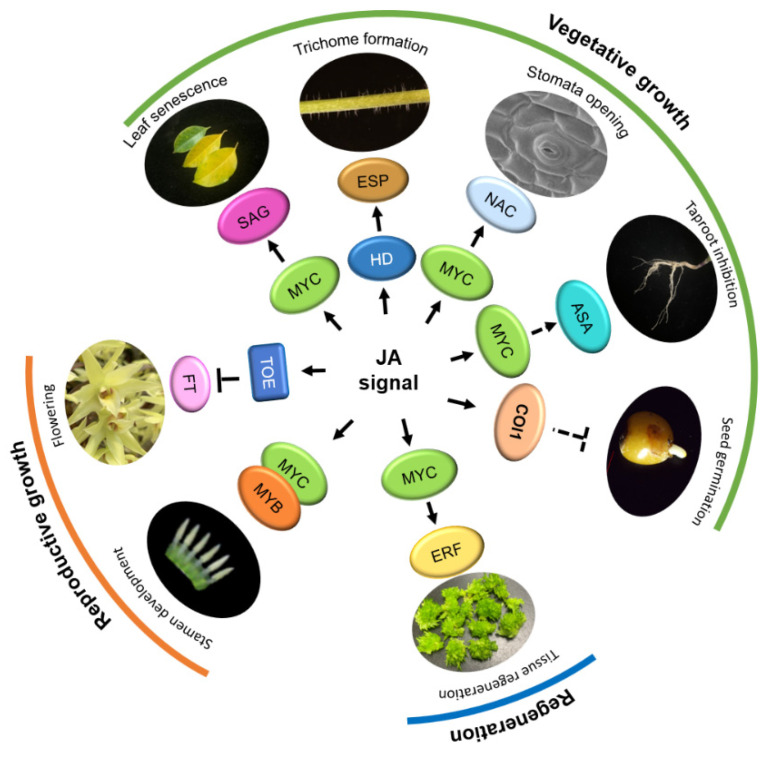
The function of Jasmonic acid (JA) in plant growth and development. In the reproductive growth stage, bHLH IIIe transcription factors (TFs), MYCs, and MYBs promote stamen development. JA interacts with AP2 family TOEs to inhibit FT transcription and regulate plant flowering. In the vegetative growth stage, MYCs and ASA jointly inhibit the elongation of the taproot; MYCs interact with SAG to activate JA-induced leaf senescence; MYC and downstream NAC TFs promote stomatal opening; HD-ZIP family members (HDs) activate the expression of expansion proteins gene (*ESP*) and regulate trichome formation; JA also inhibits seed germination through COI1. In addition, MYC and ERF jointly promote tissue regeneration. Arrows: Activations; Bar-headed arrows: Repressions.

**Figure 4 ijms-23-03945-f004:**
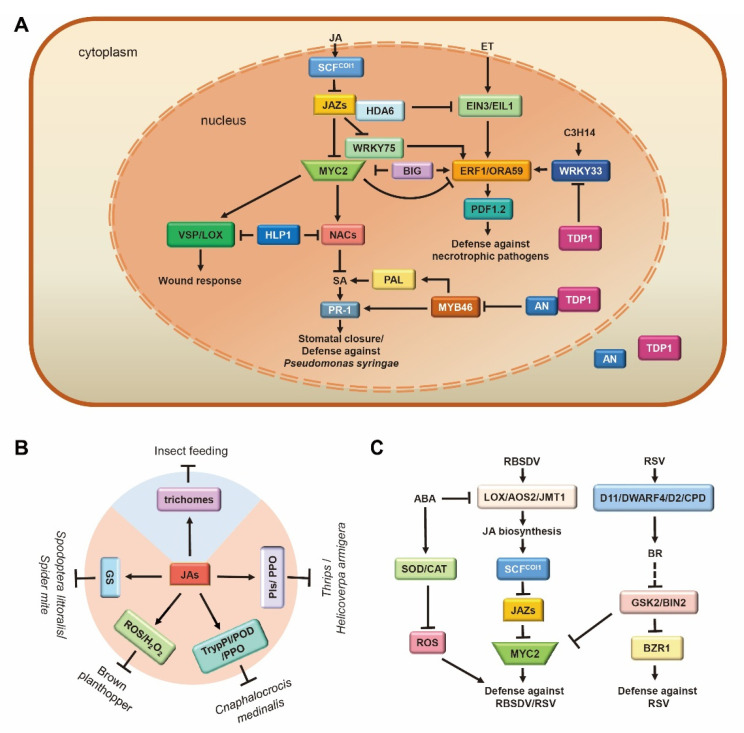
Multi-functional role of jasmonic acid (JA) interaction module in stress. (**A**) JA-ET-SA network modulates the defense against pathogens in Arabidopsis. JA and ET synergistically regulate the defense against necrotrophic pathogens. *BIG* gene positively/negatively regulates MYC2/ERF1 and coordinates the defense of plants against pathogens and insects. NACs transcription factors in the JA pathway and SA pathway antagonize the regulation of stomatal dynamics and defense response against *P. syringae*. (**B**) JA defense against herbivorous insects through physical (blue) and chemical (orange) means. JA promotes the accumulation of GS (defense against *Spodoptera littoralis*/Spider mite), ROS/H2O2 (defense against brown planthopper), TrypPI/POD/PPO (defense against *Cnaphalocrocis medinalis*), and PIs/PPO (defense against *Thrips*/*Helicoverpa armigera*), as well as the initiation of trichomes to defend against a variety of herbivorous insects. (**C**) JA-BR-ABA crosstalk-regulated defense response to viruses. JA and BR synergistically defend against RSV through MYC2/BZR1. ABA antagonistically regulates the defense response to RBSDV by inhibiting the production of ROS and the synthesis of JA. Arrows: Activations; Bar-headed arrows: Repressions. JA and ET are two crucial plant hormones that co-operate to activate defenses against necrotrophic pathogens, whereas the SA pathway triggers defenses against biotrophic and hemi-biotrophic pathogens.

**Figure 5 ijms-23-03945-f005:**
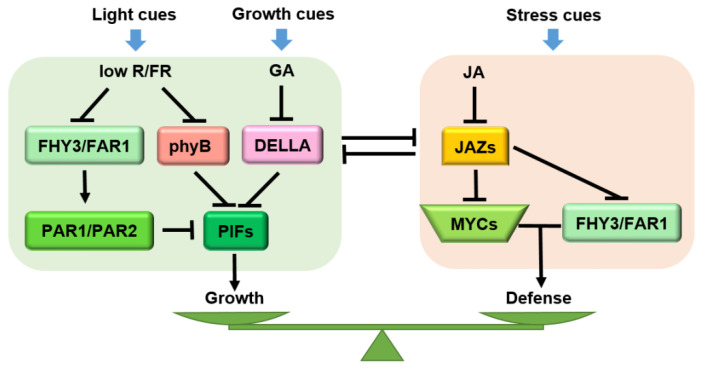
Molecular mechanism of JA-GA-phyB crosstalk that governs growth and defense. When defense cues are generated, JA dominates the defense response and regulates the growth-defense balance together with GA/phyB. GA and phyB jointly inhibit plant growth under shade conditions, while JA and GA antagonize and regulate the plant growth response under growth status. Arrows: activations; bar-headed arrows: repressions; lines: interactions.

**Table 1 ijms-23-03945-t001:** Mutants and overexpression lines of JA biosynthesis pathways and core signaling components in Arabidopsis, *Oryza sativa*, and *Solanum lycopersicum*.

Species	Name	Description	Alteration in JA Responses	Ref.
*At*	*fad3-2fad7-2fad8*	Cross between *fad3-2*, *fad7-2* and *fad8*	No JA produced; Male sterile; Hypersensitive to *Alternaria brassiccola*	[46,47]
	*opr3*	T-DNA insertion mutant	No JA produced; Defective anther and pollen development, male sterile; Enlarged petals; Resistant to *A. brassicicola* infection	[47,48]
	*jar1-1*	/	JA-insensitive phenotype; Defect in JA-Ile synthesis; Increased susceptibility to *P. irregular*	[49,50]
	*jassy*	T-DNA insertion mutant	Defective in OPDA transportation; Reduced cold tolerance; Increased susceptibility to *B. cinerea*	[22]
	*dad1*	T-DNA insertion mutant	Defective in JA biosynthesis; Male sterile	[51]
	*dde1*	T-DNA insertion mutant	Defective in JA biosynthesis; Male sterile	[52]
	*aos*	T-DNA knockout mutant	No JA produced; Male sterile	[53]
	*acx-1acx-5*	Cross between *acx-1* and *acx-5*	Poor pollen viability; Increased susceptibility to *Trichoplusia ni* larvae and *Frankliniella occidentalis*; Remain resistant to the *A. brassicicola*	[54]
	*lox3lox4*	T-DNA insertion mutant	Defective in JA biosynthesis; Male sterile	[55]
	*aim1*	T-DNA insertion mutant	Defective in wound-induced formation of JA; Defective in floral development	[56,57]
	*pex6*	/	Defective in wound-induced formation of JA; Increased OPDA level	[56]
	*kat2kat5*	Knockout mutant	Growth defect; Male sterile; Phenotype similar to *aim1*	[58]
	*cyp94c1−1*	T-DNA insertion mutant	Decreased 12COOH-JA-Ile accumulation	[59]
	*cyp94b3cyp94c1*	/	Increased JA-Ile accumulation; JA-insensitive phenotype; Sensitive to exogenous JA	[59]
	*coi1-1*	EMS mutagenized, W467 * nonsense mutation	JA insensitivity; Male sterile; Increased susceptibility to fungal pathogens and *Erwinea carotovora*; Dark-induced senescence	[60,61,62]
	*coi1-2*	EMS mutagenized, L245F missense mutation	Reduced JA insensitivity; Partial fertility	[27,63,64]
	*coi1-8*	EMS mutagenized, G543L missense mutation	Reduced JA insensitivity; Insensitivity to JA-inhibitory root elongation; Partial fertility	[27]
	*coi1-16*	/	Fertility in a temperature-sensitive manner	[28,65]
	*coi1-20*	EMS mutagenized	Male sterility; Resistant to *P. syringae*	[60]
	*coi1-21*	T-DNA in F-box	Male sterility	[65,66]
	*coi1-30*	/	Enhanced resistance to *P. syringae*; Longer hypocotyls and petioles under low-intensity light conditions; Early flowering	[67,68]
	*coi1-37*	T-DNA insertion line, a 1537 bp deletion in the promoter, and the first exon	Male sterility; Leaf epinasty; Dark green leaves; Strong apical dominance; Enhanced meristem longevity	[66]
	*jaz1-1*	Loss-of-function	Normal JA responses	[69]
	*jaz2-1*	T-DNA insertion at the fourth intron	JA insensitivity	[70]
	*jaz2-3*	Transposon insertion leads to JAZ2 knockout	Partially impaired in stomatal closing; More susceptible to *P. syringae*	[67]
	*jaz2∆jas*	T-DNA insertion in the third exon; Lack the Jas domain	Resistant to *P. syringae* and necrotrophs	[67]
	*jaz5-1*	/	Normal JA responses	[69]
	*jaz6-1*	/	Short filament; Delayed anther dehiscence; Unviable pollen grains	[71]
	*jaz7*	T-DNA insertion at the promoter, overexpression	JA sensitivity; Significantly short roots; Reduced weight; Enhanced defense	[36,70]
	*jaz7-1*	T-DNA insertion at 384 bp from the 5’-UTR, Loss-of-function	Week regulation of cambium initiation; Dark-induced leaf senescence hypersensitive	[43,72,73]
	*jaz9-1*	Loss-of-function	Partial GA insensitivity; Normal JA phenotype	[68]
	*jaz9-3*	Loss-of-function	Partial GA insensitivity	[68]
	*jaz10-1*	Open reading frame was disrupted, Loss-of-function	JA-hypersensitive; Enhanced susceptibility to *P. syringae* infection; Enhanced cambium initiation	[69,72,74]
	*jaz10-2*	A weak allele, Loss-of-function unclear	Weak regulation of cambium initiation	[72]
	*jazQ*	T-DNA insertion mutations in 5 *JAZ* genes (*JAZ1/3/4/9/10*)	JA-hypersensitive root growth; Enhanced susceptibility to *P. syringae*; Heightened resistance to *Trichoplusia ni*	[13]
	*jazD*	T-DNA insertion mutations in 10 *JAZ* genes (*JAZ1-7, -9, -10, -13*)	Resistant to insect herbivores and fungal pathogens; Slow vegetative growth; Poor fertility	[43]
	*myc2-3*	/	Reduced formation of interfascicular cambium	[72]
	*myc3-1*	*MYC3* knockout	Enhanced resistance to *P. syringae* and *S. littoralis* larvae	[32]
	*myc4-1*	*MYC4* knockout	Enhanced resistance to *P. syringae* and *S. littoralis* larvae	[32]
	*myc5*	/	Normal development in flower and stamen	[75]
	*myc1/3/4*	/	Hypersensitive to *S. littoralis* and spider mite; reduced JA-mediated root inhibition	[76,77]
	*myc2/3/4/5*	/	Short filament; Delayed anther dehiscence; Unviable pollen grains	[75]
	OE *JAZ1∆3A*	Lack residues 202-228	JA-insensitive phenotypes; Male sterility	[29]
	OE *JAZ8*	/	JA-insensitive root growth; Vulnerability to herbivore attack	[36]
	OE *JAZ7*	/	Enhanced drought tolerance	[78]
	OE *JAZ9*	/	Longer hypocotyls and petioles under low-in-tensity light condition; Early flowering	[29]
	OE *JAZ10.4*	Lack the Jas domain	JA-insensitive; Resistant to JA-induced degradation	[79]
*Os*	*aoc-2*	T-DNA insertion mutant	Decreased JA accumulation; Susceptible to BPH attack	[80]
	*cpm2*	An 11 bp deletion within the first exon of AOC	Enhanced adaptability to drought; Male sterile; Strong root systems	[81]
	OE *JMT*	/	Increased MeJA accumulation; Reduced height and yield; Increased resistance to BPH nymphs	[82]
	*coi1-13*	RNAi line	JA-insensitive; Increased plant height; More susceptible to virus infection	[68,83]
	*coi1-18*	RNAi line	JA-insensitive; Increased plant height	[68]
	*jaz1*	T-DNA insertion mutant	Increased drought tolerance	[84]
	*jaz6*	/	Normal JA responses	[85]
	*myc2*	Loss-of-function	Reduced JA-mediate RSV defense response	[86,87]
	*OE JAZ1*	/	More sensitive to drought stress	[84]
	*OE JAZ6*	/	JA-insensitive phenotype; Abnormal spikelet development; Weak root inhibition	[85]
	*OE JAZ8∆C*	Lack the Jas domain	JA-insensitive phenotype; Negatively regulated the JA-induced resistance to *Xoo*	[88]
	*OE* *JAZ13a*	Lack an intron	JA-insensitive root growth; Developed lesion mimics in the sheath and tillers	[89]
*Sl*	*def1*	/	Decreased JA accumulation; Increased susceptibility to *Manduca sexta*	[90,91]
	*spr1*	/	Decreased JA accumulation; Defective in wound signal-mediated PI expression	[92]
	*spr2*	/	Defective in JA biosynthesis; Increased susceptibility to tobacco hornworm larvae	[93,94]
	*jai1*	A 525 bp downstream intron-1 sequence deletion	Reduced pollen viability; Abnormal development of glandular trichomes; Increased susceptibility to two-spotted spider mites, *B. cinerea*, *Pythium*, and *Fusarium*	[95,96,97]
	*JAZ2Δjas*	Lack the Jas domain	Inhibited stomatal reopening by COR and enhanced resistance to *P. syringae*; Remain resistant to the *B. cinerea*	[98]
	OE *JAZ2*	/	Quicker leaf initiation; Reduced plant height; Decreased trichomes; Earlier lateral bud emergence; Advanced flowering transition	[99]

## Data Availability

Not applicable.

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
