# Peer review of "Jasmonate Signaling Pathway Modulates Plant Defense, Growth, and Their Trade-Offs"

_ijms, 2022, doi:10.3390/ijms23073945_

Round 1

Reviewer 1 Report

In the present manuscript entitled “Jasmonate signaling pathway modulates plant, defense, growth, and their trade-offs” the authors present a review summarizing the current knowledge on the biosynthesis, signaling, and role of jasmonic acid (JA) and its derivatives in plants.  A special emphasis was put on the crosstalk between JA and the growth hormone gibberellin, as well as JA and photosensitive signaling pathways to regulate plant growth-defense trade-off. While I think the manuscript represents a huge effort to summarize all the advances in JA research, my criticisms are listed below.

Major Comments:

  • In general, the manuscript is well written. However, the work suffers from some general language and grammar issues. Moreover, the writing style needs thorough polishing to improve the flow of the paper and legibility. I recommend the authors use formal English and appropriate scientific expressions, avoiding expressions like the one found in lines 114-115: “When plants encounter adversities, they will biosynthesis JA-Ile as soon as possible”
  • The information detailed in sections 2 and 3 is mostly published in Li et al. Metabolism, signaling, and transport of jasmonates, Plant Communications (2021), https://doi.org/10.1016/. Therefore, I advise the authors to change the focus of this article. In my opinion, it would be interesting to summarize the state of knowledge regarding the role of JA in plant growth and defense trade-off.
  • The article is not well balanced among different parts and most sections contain redundant information. For instance, sections 3 and 4 can be integrated together.
  • On the other hand, there are many errors concerning abbreviations. Furthermore, species nomenclature and plant mutants' names should be written in italics, please review accordingly.
  • I found table 1 and table 2 very interesting, however, the description is missing in some of the mutants.
  • The title proposed by the authors does not summarize what is described in the manuscript and it should be changed.

Minor Comments

  • In the introduction, paragraphs 1 and 2 contain redundant information. In my opinion, would be adequate to reconsider re-writing this section to make the article more readily accessible.
  • Line 24: “MeJA” this abbreviation needs to be explained.
  • Line 30: As a suggestion, change “Much work has been invented to…” to “During the last decades, much work has been done to investigate…”.
  • Line 32: Change “An appropriate level of JA in essential for inhibiting root growth, stimulating flower organ development, and promoting the leaf senescence …” to “Augmentation of the endogenous JA levels provoke root growth inhibition, stimulate flower organ development, and promote lead senescence”.
  • Line 34: Re-phrase “Furthermore, excessive JA accumulation in plants will cause excessive defense and affect its growth” to “Furthermore, excessive JA accumulation would trigger an overactivation of the defense machinery, which in turns comes at the expenses of plant growth”
  • Lines 39-41: To me, it is not clear how the authors use the references to support different statements.
  • Lines 53-54: “researchers have made a great job on how JA, a key signal molecule, is produced and accumulated”, personally, I believe that authors should be careful when providing a single reference to support these kinds of statements. When illustrating the great effort that the scientific community made in unraveling certain questions, several literature references must be given. This will provide a solid argumentation of the mentioned information. Nevertheless, In my opinion, this sentence is unnecessary and can be removed.
  • Line 55: “Linoleic acid” needs revision.
  • Lines 77: Re-phrase “Catalyze JA to the receptor”.
  • Line 79: “the bioactive CORONATINE INSENSITIVE1 (COI1)-JAZ ligand”, signaling pathway is explained in section 3, thereby I suggest rewriting this sentence.
  • Line 103: Change “JASMONATE-ZIM DOMAIN PROTEIN (JAZ)” to “JASMONATE-ZIM DOMAIN (JAZ) proteins”
  • Lines 128-129: This sentence needs revision.
  • Lines 172: “JAZ deficiency in Marchantia maybe as a consequence of constitutive activation of MYC TFs, which needs to be confirmed further” needs to be rewritten.
  • Lines 261-266: The information provided in this paragraph was already explained in previous sections.
  • Section 5.1: References supporting information are missing.
  • Figure 3: It would be helpful to have the specific transcription factors controlled by JA, not only in the figure legend but also in the figure itself.
  • Lines 374 to 391: It has been demonstrated that trichome development is regulated by auxin e.g. Zhang et al. Plant and Cell Physiology (2015) https://doi.org/10.1093/pcp/pcv136. In addition, several works point towards a JA-dependent indole-3-acetic acid biosynthesis to control plant growth i.e. Hentrich et al. Plant Journal (2013) https://doi.org/10.1111/tpj.12152;  Machado et al., Plant Physiology (2016) https://doi.org/10.1104/pp.16.00940; and Pérez-Alonso et al. international journal of molecular sciences (2021)  https://doi.org/10.3390/ijms22189768. Therefore, I think it would be worth it to consider a section reviewing the relation between JA and indole-3-acetic acid.
  • Section 6: To my understanding, this section is chaotic and lacks the reading flow. Please re-write.

Reviewer 2 Report

At present, jasmonates are, probably, the most studied phytohormones, and a big number of reviews are devoted to the analysis of modern literature on this topic, including those describing the role of jasmonates in growth-defense tradeoffs. Therefore, rather high requirements are imposed on a review on this topic, especially on its novelty. In this review, the authors have tried to present a fairly complete analysis of the literature data on this topic, with consideration of the molecular mechanisms underlying the jasmonates function implementation and signaling pathways.

The two main shortcomings in this review are (1) the presence of information previously published in other reviews and (2) an inaccurate use of scientific language and terms. The manuscript requires careful English correction. In addition, the authors do not touch upon the aspect of the jasmonate-regulated redistribution of metabolic fluxes, although it is obvious that the metabolic fluxes primarily determine both the processes associated with growth and productivity and the biosynthesis of protective compounds. Example: Jasmonates-Mediated Rewiring of Central Metabolism Regulates Adaptive Responses doi.org/10.1093/pcp/pcz181. Only at the end of the manuscript the authors have acknowledged the importance of this aspect and have mentioned: “it is possible that the growth-defense tradeoffs cannot rely on transcriptional networks only but also depends on strong metabolic constraints owning to the reallocation of metabolites to defense”, but did not discuss the topic thoroughly.

For publication in such a high-ranking journal as IJMS it is necessary to significantly change the content of the review. Less attention has to be given to the subjects already widely presented in the other review articles, such as jasmonates biosynthesis and components of the hormone signal transduction pathway, and the review has to be enriched with new data. 

For review articles, good English and presentation of the text are even more important than for experimental articles, so the language should be corrected and special attention paid to the use of scientific terms. Some examples of poor text presentation are given below.

Line 39 It is written: “with other plant biological processes, such as salicylic acid (SA), ethylene (ET), gibberellin (GA), photosensitive signal pathway, and so on”. Do authors mean other signaling pathways, not processes?

Line 131 “leads to the insensitivity of JA” Should it be “leads to the insensitivity to JA”?

“PUFAs initiate JAs formation” – Does it mean “PUFAs are the substrates for JA biosynthesis”?

In several places in the text “terminal mechanisms of JA signals” is used. Do authors mean jasmonates catabolism? This terminology is used in other alticles, for example, in the article (Zhu, T., Herrfurth, C., Xin, M. et al. Warm temperature triggers JOX and ST2A-mediated jasmonate catabolism to promote plant growth. Nat Commun 12, 4804 (2021). https://doi.org/10.1038/s41467-021-24883-2)

“Galactolipids release α-linolenic acid (18:3) by phosphatase A1 (PLA1)” – Phospholipase? Phospholipids or galactolipids?

“including binding amino acids” - binding or conjugation

It is mentioned: “OPDA in chloroplasts is transported to the peroxisomes via the peroxisomal ABC-transporter1 (PXA1)”. Should be “from chloroplasts” and, importantly, recently discovered chloroplast envelope-localized transporters are not mentioned.

There are many technical errors in the text. Only several examples are below:

“Arabidopsis myc234 triple mutant” – myc 2/3/4

“MpJAZ transcripts exist a splicing (elimination of the Jas domain) in response to wounding” – the sentence is not correct

  “we presents JA biosynthesis mechanism, a core signaling pathway” – Please, remove “mechanism”, should be “we present”, without “s” at the end

“photosensitive signal pathways to” – light-sensing?

Line 48 “closely relationship” – close relationship

Line 55 please, correct “alpha-linolenic acid”

“Peroxidasome” – peroxisome

Latin names of plant and pathogen species are not italicized, examples are on line 304: Oryza sativa, and Solanum lycopersicum

Line 246 26s – should be 26S

In the presented conclusion, part of the text is repeated in the other chapters of the review, and part – is not the essence of the discussed topic, such as species-specific differences in jasmonate-based system, or JAs-regulated synthesis of secondary metabolites in medicinal herbs. I would suggest leaving only the text summarizing the information about the growth-defense trade-off.

Round 2

Reviewer 1 Report

Firstly, I would like to congratulate the authors for the significant improvement of the MS. However, there are some issues that in my opinion, need to be addressed.

General comments:

The MS still contains errors concerning Latin names of plant and pathogen species, as well as genes and mutants’ names, they are not italicized, please review accordingly. Besides, I still find an inaccurate use of scientific expressions and terminology. Along the presented work, authors described the interaction of JA with multiple hormones, however, at the end of the manuscript they focused on the crosstalk of JA with Gibberellins (Section 7.2). If the purpose of the section is to highlight the role of JA in mediating growth-defense trade-offs, the relation between auxin and oxylipins must be discussed thoroughly (doi: 10.1111/j.1744-7909.2011.01053.x.).

1.- Line 226: It is written “The COI1 protein is strictly regulated by a dynamic balance of SCFCOI1….” I assume that this is the consequence of a decrease of the bioactive JA levels, it´s that right? To make section 3.2 comprehensive easier for the readers I recommend explaining the regulation of JA homeostasis and the termination of JA signaling in two separate sections.

2.- I suggest the authors incorporate Section 4 “Evolutionary origin of JA signaling” at the end of section 3.2 “the core JA-Ile pathway”.

3.- Lines 431 to 439: I am missing references supporting the provided information.

4.-Lines 474 to 475: Change “researchers have also uncovered other roles for JA in developmental and growth-related processes. coi1-37…” to “As described in Table 1, researchers have also uncovered other roles for JA in developmental and growth-related processes. For example, coi1-37…”.

5.- Line 481: “Functions of the JA pathway in stress response” should be “Role of JA during plant defense responses”

6.-I strongly believe that Table 1 and 2 can be merged into a single table.

7.- Figure 1 should be referred to in section 3.1, and question marks need to be explained. I would appreciate it if the authors indicated how many COI, JAZ, and MYC proteins had been described in Arabidopsis.

8.- Line 648: The information provided here is not true.  Upon attack, JA-signaling is activated, and subsequently, MYC2, MYC3, and MYC4 mediate indole-3-acetic acid biosynthesis through the activation of YUCCA9 expression.

9.- Titles from sections 6.2 and 6.3 are practically identical, I encourage the authors to make titles more attractive and informative.

10.- From my point of view section, 7 is redundant to section 5.

Reviewer 2 Report

The manuscript has improved significantly. The diagrams and informative tables will be highly appreciated by readers.

Minor comments:

  • Since more than one lipase contributes to JA biosynthesis, authors can use the more general enzyme term “lipase”, or they should list all lipases involved in JA biosynthesis, phospholipase and glycerolipid lipase. And, in any case, it should be written “phospholipase”, not “phosphatase”. Here is the fragment from the articles regarding the lipases involved in JA synthesis (https://doi.org/10.1104/pp.110.155093): “The DEFECTIVE IN ANTHER DEHISCENCE1 gene (DAD1) has been shown to encode a chloroplastic glycerolipid lipase. Initially, DAD1 was characterized as a phospholipase (PLA) A1 involved in JA biosynthesis in Arabidopsis flowers (Ishiguro et al., 2001)”.
  • The included reference 42 (doi: 10.1093/pcp/pcz181) would be appropriate after the text about metabolites reallocation “also depends on strong metabolic constraints, due to the reallocation of metabolites to for defense”, but does not fit to support the conclusion about JA catabolism (lines 220-224).
  • Only as a suggestion: In the most places of the text, the term “tolerance” would be more suitable than the term “resistance”.
